# Coal Ash Content Measurement Based on Pseudo-Dual Energy X-ray Transmission

**Xiufeng Zhang [1,2], Long Liang [1], Taiyou Li [2], Jiakun Tan [1,\*], Xingguo Liang [2] and Guangyuan Xie [1]**

1    School of Chemical Engineering and Technology, China University of Mining and Technology, Xuzhou 221116, China; xiufeng_smile@163.com (X.Z.); lianglong1218@sina.com (L.L.); xieguangyuan1962@sina.com (G.X.)
2    Tianjin Meiteng Technology Co., Ltd., Tianjin 300110, China; cgsirc@163.com (T.L.); 18649004334@163.com (X.L.)
\*    Correspondence: JiakunTan@cumt.edu.cn

**Abstract:** The real-time ash content measurement is the fundamental condition for the timely adjustment and intelligent control of operation parameters in coal production and utilization industry. In the present work, a real-time ash content analyzer based on the pseudo-dual energy X-ray transmission was developed. The feasibility of this X-ray ash content analyzer was validated by the linear relationship between ash content and five characteristic parameters of X-ray. The conditions of wave filter, tube voltage, and tube current were optimized. The comparison between the ash contents measured by muffle furnace and the X-ray ash content analyzer was conducted in laboratory and industry. It was found that the absolute error was smaller than 1% for clean coal with ash content of approximately 10%, and the possibility of the absolute error smaller than 0.5% was higher than 85%.

**Keywords:** ash content; real-time measurement; X-ray; coal

## 1. Introduction

Coal is an important energy and chemical engineering resource in the world [1]. The reasonable development and utilization of coal resource has meaningful significance to the economy and environment of the world, especially for the countries that have large production and consumption of coal, such as China. Ash content is a key evaluation parameter of coal quality, as well as a standard that guides the reasonable utilization of coal. The conventional ash content measurement is to heat the coal in muffle furnace at a temperature higher than 800 °C, which causes high consumption of energy and time. Additionally, timely adjustment of operation parameters in coal preparation plant could not be achieved due to the delay of ash content result that was usually obtained after several hours [2]. Therefore, it is important for coal industry researchers to seek real-time ash content measurement techniques.

The available real-time ash content measurement techniques include radiometry, photoelectric measurement, and image processing [3–5]. Among these techniques, radiometry is used most widely in coal industry, which can be divided into two categories, i.e., source-based radiometry, and passive radiometry.

Source-based radiometry is commonly based on the reflection and scattering of low energy γ-ray, the annihilation radiation of high energy γ-ray, and the transmission of dual energy γ-ray [6–9]. The reflection and scattering of low energy γ-ray requires complicated measurement conditions, which are difficult to satisfy in industry. The annihilation radiation of high energy γ-ray is usually interfered by the high atomic number element and heavy medium that remain on coal. Thus, the popular γ-ray ash content analyzer in market is mainly based on the principle of transmission of dual energy γ-ray [10–13]. It uses the low energy 241 Am and the decay of 137 Cs to calculate the ash content.

In passive radiometry, ash content is calculated according to its relationship with the released γ-ray of the radioactive elements (K, Rb, Th, U, etc.) embedded in coal during

their decaying process [14]. The main drawback of this technique is that the radioactive elements in sediment sand rock are so rare that is not detectable. Therefore, the application of passive radiometry is limited to the coal that the mineral content has good correlation with the released γ-ray from the radioactive elements [14].

Besides the above-mentioned techniques, neutron activation is also used in real-time ash content measurement, but with fewer applications [15–17]. The working principle of neutron activation is to identify and quantify the element in coal by the inelastic scattering and captured radiation of neutron [18,19]. In recent years, the real-time ash content measurement sensors based on induced laser [1] and X-ray fluorescence have also been developed [20]. The laser-induced breakdown spectroscopy technique has been rapidly developed recently. Based on this technique, Body and Chadwich [21] invented the coal quality analyzer, and Gaft [22,23], Matoe [24], and Aurelio [25] validated the feasibility and accuracy of the laser-induced breakdown spectroscopy in real-time ash content measurement. However, the ash content analyzers based on neutron activation and laser-induced breakdown spectroscopy are expensive and complicated in operation. The price of neutron activation and laser-induced ash content analyzers are respectively higher than USD 780,000 and USD 390,000. It costs approximately USD 160,000 for an X-ray ash content analyzer. The neutron activation ash content analyzer includes radioactive reactor that requires strict control procedures and special maintenance, while laser-induced ash content analyzer is usually coupled with sampling and preparation system and other auxiliary equipment.

X-rays are an electromagnetic wave with a wavelength between 0.001 to 10 nm and photonic energy between 120 and $1.2 \times 10^6$ eV. X-rays are emitted through the collision between the anode target and the thermal radiation electrons from the cathode of an X-ray tube under electrified condition. During the collision process, energy is released due to the reduction of electron velocity or electronic transition, resulting in the formation of bremsstrahlung and characteristic radiation. Due to the rapid development and wide application of X-ray techniques in coal and mining industry, the separation of coal and gangue based on X-ray transmission has been achieved and is quickly becoming increasingly popular [26]. The basic principle of this technology is that the decay degree of X-rays varies with the materials of different density, so there is significant difference of X-ray intensity after X-rays are emitted to coal or gangue particles. The analysis of ash content is based on the X-ray pattern, which is obtained from the electric signal transformed from the decayed X-ray intensity detected by the X-ray receiver [27,28].

This work presents a novel real-time ash content measurement technique based on pseudo-dual energy X-ray transmission. The so-called pseudo-dual energy means that the X-ray scintillation receiver is composed of two layers of silicon photodiode array composed of different scintillation materials and copper wave filter, thus the received X-ray can be divided into high energy and low energy ranges. The pseudo-dual energy X-ray transmission has been used in coal separation and gangue disposal [28,29]; however, its application in coal ash content measurement has not been studied. The relationship between ash content and characteristic parameters of pseudo-dual energy X-ray was investigated and the conditions of wave filter, tube voltage, and tube current were optimized. Finally, the absolute error and standard deviation between the ash contents measured by pseudo-dual energy X-ray ash content analyzer and muffle furnace were compared.

## 2. Materials and Methods

### 2.1. Coal Sample

The coal sample used in this work was collected from the run-of-mine coal (−50 mm) in Xiegou coal mine, Shanxi Coking Coal Energy Group, China. The clean coal with density lower than 1.4 g/cm$^3$ was obtained from float-and-sink experiment, and was crushed and ground to −74 μm to conduct the proximate analysis, ultimate analysis, and X-ray fluorescence (XRF) measurement (S8 TIGER, Bruker, Karlsruhe, Germany). For proximate analysis, the run-of-mine coal was also measured. The results of proximate analysis, ultimate analysis, and XRF are shown in Tables 1–3, respectively. Tables 2 and 3 shows that

there were aluminum and silicon elements in the clean coal sample, which may come from the silicate and aluminosilicate minerals. The carbon element in Table 3 represents the light element such as H, C, N, etc., which is also called combustible matter, mainly composed of the organic matter and crystal water in the sample.

**Table 1.** The proximate analysis of the coal sample (air-dried basis).

| Sample | Moisture (%) | Ash Content (%) | Volatile Matter (%) | Fixed Carbon (%) |
|---|---|---|---|---|
| Clean coal | 3.14 | 10.58 | 28.33 | 54.48 |
| Run-of-mine coal | 3.86 | 19.31 | 27.24 | 55.50 |

**Table 2.** The ultimate analysis of the clean coal sample (air-dried basis).

| Element | C | H | O | N | S |
|---|---|---|---|---|---|
| Concentration (%) | 81.32 | 5.22 | 10.59 | 2.06 | 0.81 |

**Table 3.** Element composition of the clean coal measured by XRF. Conc.—Concentration.

| Element | $SiO_2$ | $Al_2O_3$ | $CO_2$ | $SO_3$ | CaO | $Fe_2O_3$ | $TiO_2$ | Cl | $P_2O_5$ | MgO | $K_2O$ |
|---|---|---|---|---|---|---|---|---|---|---|---|
| Conc. (%) | 5.61 | 5.55 | 84.00 | 3.26 | 0.692 | 0.295 | 0.223 | 0.192 | 0.081 | 0.056 | 0.0388 |

The composition of macroscopic coal rock and maceral was also analyzed by the microphotometer (DMRXP-MPV-SP, LEICA, Wetzlar, Germany). The run-of-mine coal was crushed to −1 mm and was then sieved. The coal particles with particle size of 1–0.1 mm were used in the macroscopic coal rock and maceral composition measurement. Before measurement, the coal particles were rinsed with deionized water, and a polished section was prepared after drying. The polished section was rinsed by ethanol and attached on a sample holder after drying. Oil immersion objective was used in the measurement.

The macroscopic coal rock measurements indicated that the coal sample was mainly composed of clarain, and then durain and vitrain. Fusain was only found in the pores in some individual parts. The maceral measurements show that the main organic microscopic compositions were vitrinite and inertinite. The proportion of vitrinite was between 34% and 56% and averaged at 41%, while it was between 24% and 42% and averaged at 36% for inertinite. The minerals were mainly the clays with disseminated, crumby, schistose, and filling structure. Some calcite and pyrite were also found as scattered particles or veined fillings in the organic base. Figure 1 shows the microscope images of some typical minerals distributed in coal sample.

*2.2. Set up of the X-ray Ash Content Analyzer*

The working principle of X-ray ash content analyzer is the Lambert–Beer law as shown in Equation (1):

$$I = I_0 e^{-\mu T} \tag{1}$$

where $I_0$ and $I$ are the intensity of X-ray before and after the X-ray passes through the detected sample, $\mu$ is the mass adsorption factor, and $T$ is the thickness of detected sample [11]. According to the Lambert–Beer law, there is an exponentially decaying relationship between the residue X-ray intensity after passing through a sample and the mass adsorption factor and thickness of the sample. The mass adsorption factor is closely correlated to the atom number and concentration of the elements in the sample. Therefore, if the sample thickness is constant, the density and ash content of the sample could be calculated based on the intensity decaying extent of X-ray with high and low energy after passing through the sample. For this reason, a sample flattening system was coupled with the X-ray ash content analyzer to control and measure the sample thickness, which was inserted into the calculation model of ash content.

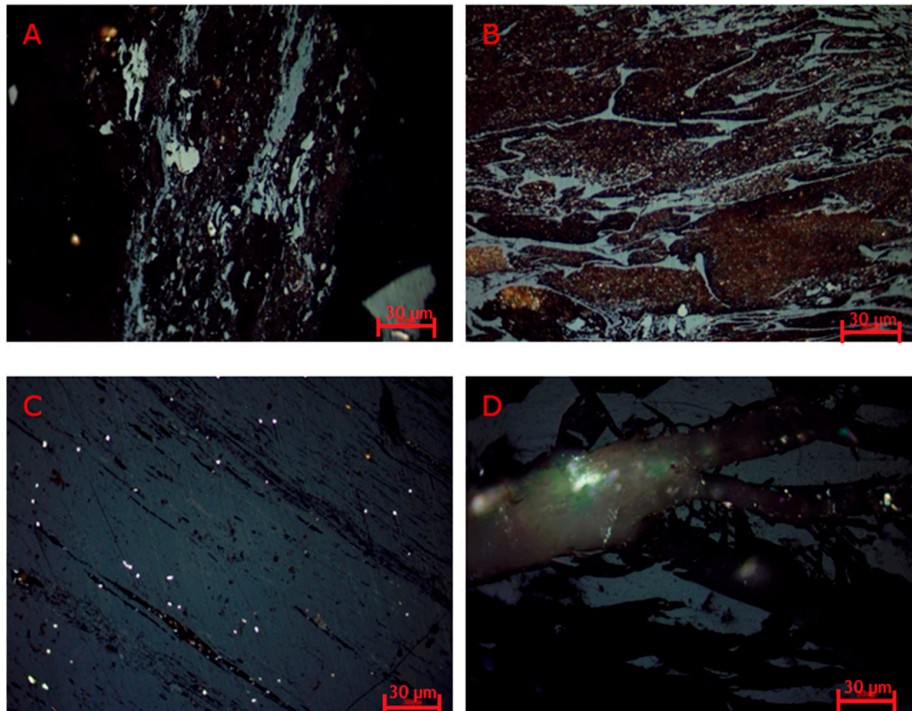

**Figure 1.** Microscope image of the minerals in coal sample: (**A**) grey clay glues with organic fragments; (**B**) grey clay fills in telinite cell lumen; (**C**) scatter distributed bright pyrite; (**D**) bright calcite veined fills in crack.

The schematic of X-ray ash content analyzer (Tianjin Meiteng Technology Co., Ltd., Tianjin, China) is shown in Figure 2. It was composed of an X-ray emitter system, lead shield, linear array detector system, and sample flatten system. The X-ray emitter system was composed of an X-ray tube, filament transformer, high-voltage generator (maximum voltage 160 kV and maximum power 500 W), sampling circuit. All components were compacted in a metal container filled with insulating mineral oil to cool the X-ray tube. A VJ X-ray tube was used in this work. The cathode filament and anode target were both made of tungsten. The aim to use tungsten is because it can release X-rays with high energy and strong penetrability after bombardment [30]. The coal sample was delivered into the X-ray ash content analyzer by a belt conveyor. The X-ray with maximum energy of 160 KeV passed through the flattened coal sample, and was then detected by the linear array detector installed under the belt. The X-ray was received and divided into two ranges, i.e., the high energy and low energy ranges. The detected X-ray was transformed to high energy and low energy signals respectively, which were treated by the control host to calculate the ash content according to the pre-designed model. Compared to the conventional γ-ray ash content analyzer which uses dual ray origins but single sampling point, the X-ray ash content analyzer measured the sample in a fan sector, so the detection area was wide. Additionally, the X-ray ash content analyzer was coupled with machine learning algorithm, so the measurement accuracy can be improved by the self-study and algorithm optimization according to the characteristics of coal quality and the real values of ash content. The signal collecting interval of the linear array detector system was in 1~10 ms. For actual ash content measurement, the signal collected in every second was divided into eight frames to calculate the data package of ash content. The data package in every minute was averaged and reported to the user.

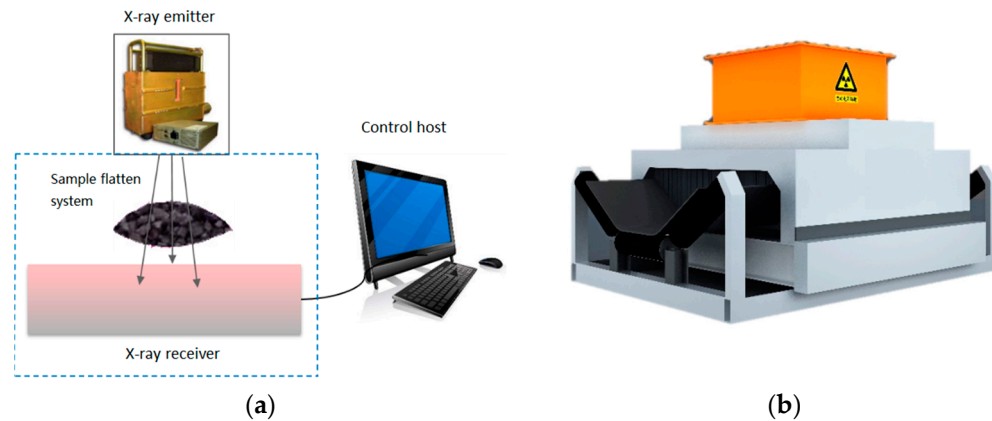

(**a**)　　　　　　　　　　　　　　　　　　　　　　　　　　(**b**)

**Figure 2.** Experimental system of the X-ray ash content analyzer. (**a**) The schematic of X-ray ash content analyzer, (**b**) Experimental system of the X-ray ash content analyzer.

## 3. Results and Discussion

### 3.1. The Relationship between Ash Content and Characteristic Parameters of X-ray

Exploration experiments were conducted to show the relationship between ash content and different characteristic parameters of X-ray. The run-of-mine coal sample was crushed to −6 mm and separated into different density fractions by float-and-sink experiments. Afterwards, the different density fractions were mixed by predetermined ratios to obtain 39 coal samples with ash contents ranging from 21% to 86%. The thickness of the coal sample during measurement was controlled to 50 mm (single layer) and 100 mm (double layer), respectively. The tube voltage was 160 kV, and the tube current was 1250 vA. No wave filter was used. All experiments were repeated three times, and the average was reported.

The characteristic parameters were calculated according to the original values of high energy and low energy that were detected when there was no sample on the belt, and the values of high energy and low energy after the X-ray passed through the sample. After preliminary comparison and screening, five characteristic parameters were selected from nine preconfigured parameters. The five characteristic parameters are residual rate of low energy, residual rate of high energy, ratio to origin, energy ratio, and decay ratio. The residue rates of low energy and high energy are referred to the ratios of the X-ray intensity in low and high energy ranges after transmission to the original X-ray intensity in corresponding energy ranges. The ratio to origin means the ratio of residue energy of X-ray to a constant, which is defined as origin. Energy ratio is referred to the ratio of the residue intensity of high energy to that of low energy. Decay ratio is ratio of the energy ratio without sample to that with sample.

The relationships between ash content measured by muffle furnace and the five characteristic parameters, as well as the linear correlation coefficients, are illustrated in Figure 3. It is shown in Figure 3 that all of the five characteristic parameters correlated well with ash content, with linear correlation factor higher than 0.9. Among them, energy ratio and decay ratio show closer correlation. Meanwhile, the results of single layer (50 mm) were higher than that of double layer (100 mm). This is because the sample of double layer adsorbs more energy than that of single layer. For the double layer, more than 90% of both the low energy and high energy would be adsorbed by the sample when the ash content is higher than 60%, so the energy that can be detected after penetration is very weak and may cause data distortion. For the single layer, the residue rate of energy was much higher, thus more accurate measurement could be achieved. With numerous exploiting experiments, it was found that the satisfactory accuracy can be obtained when the residue rate of high energy is higher than 30%.

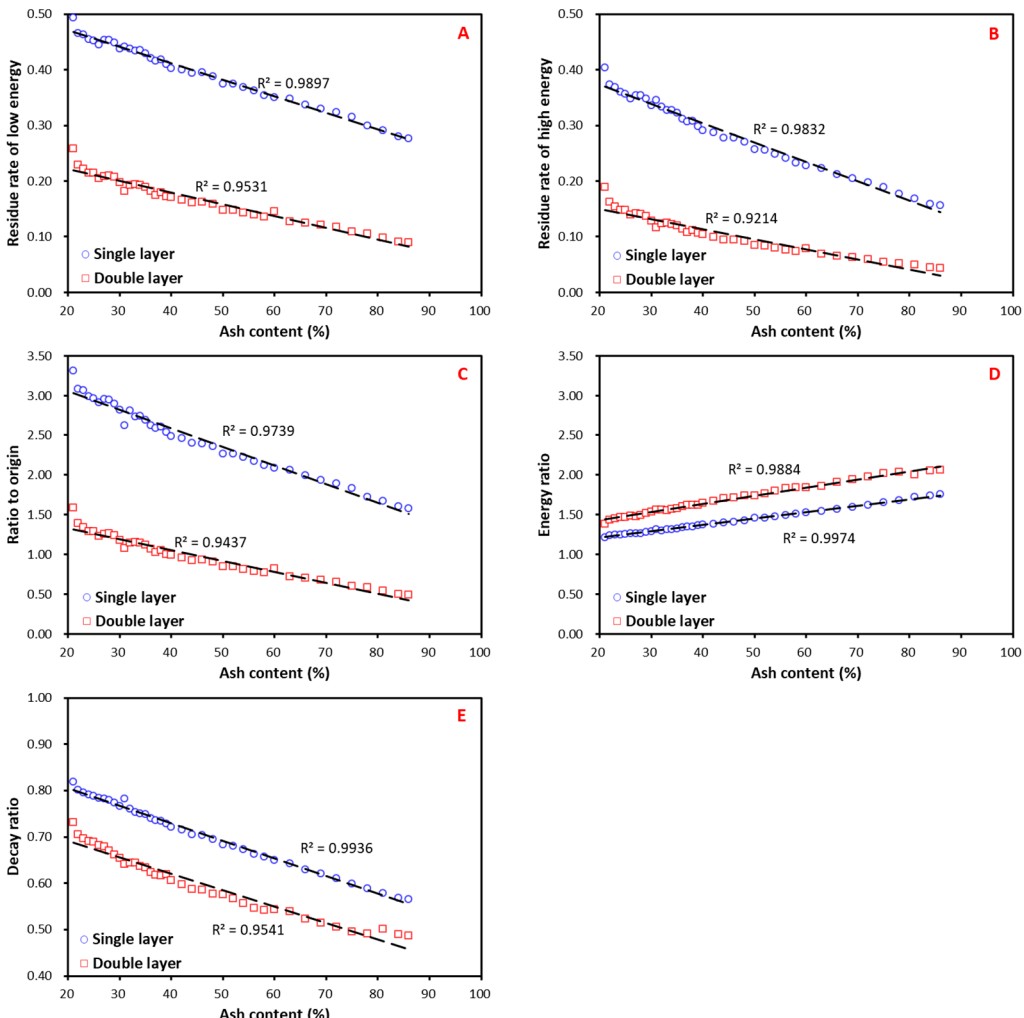

**Figure 3.** The relationship between ash content and characteristic parameters (**A**): residue rate of low energy; (**B**): residue rate of high energy; (**C**): ratio to origin; (**D**): energy ratio; (**E**): decay ratio.3.2. Optimization of Wave Filter.

The ash content measurement by X-ray requires strong penetrability of the ray, therefore the X-ray beam that is usually used has a strong intensity and wide wavelength range, as opposed to the homogeneous X-ray with single wavelength photon [31]. However, the thickness effect and beam hardening effect may occur and disturb the measurement results when X-rays with wide wavelength range pass through the sample [10]. To avoid the disturbance, a wave filter should be coupled with the X-ray emitter to narrow the wavelength range of an X-ray and reduce the soft ray in the beam. As a result, the irrelevant X-ray could be eliminated before the beam contacts the sample, and the correlation between ash content and characteristic parameters can be enhanced.

In this work, the X-ray is filtered by metal. Two kinds of wave filters made of pure nickel and ferrum-nickel alloy were investigated. After preliminary exploration, pure nickel showed better filtration performance. Subsequently, the thickness of pure nickel wave filter was studied. The thickness increased from 0.1 mm to 2.0 mm by step of 0.1 mm. A single layer (50 mm) sample was used in the experiments. The tube voltage and tube current were 160 kV and 1250 vA, respectively. All experiments were repeated six times, and the average was reported.

Overall, six coal samples with ash content of 22%, 23%, 24%, 25%, and 26% were selected from the original 39 coal samples in the exploration experiments to conduct the experiments of wave filter optimization. The reason why the samples with similar ash content were used rather than that with sharp difference is that the correlation between

characteristic parameters and ash content is usually worse when the ash content difference is smaller. If good correlation can be found under this condition, then better results can certainly be obtained when the ash content difference is large. On the other hand, the ash content analyzer is used to measure single sample in actual production process, such as run-of-mine coal, clean coal, or middlings, the ash content fluctuation of these single samples would not very large, and is usually within 5%.

The results are shown in Figure 4. It can be seen that the linear coefficient factor reduced significantly compared to that in the exploration experiments. This is mainly attributed to the decrease of the number of samples from 39 to 6. Generally, the linear coefficient factor first decreased and then increased with the increase of wave filter thickness, as shown in Figure 4. The optimum correlation between the ash content measured by muffle furnace and the five characteristic parameters was obtained with pure nickel wave filter of 2 mm. The thickness was not further increased, since with a larger thickness, the X-ray would be too decayed too to pass through the sample with moderate and high ash content.

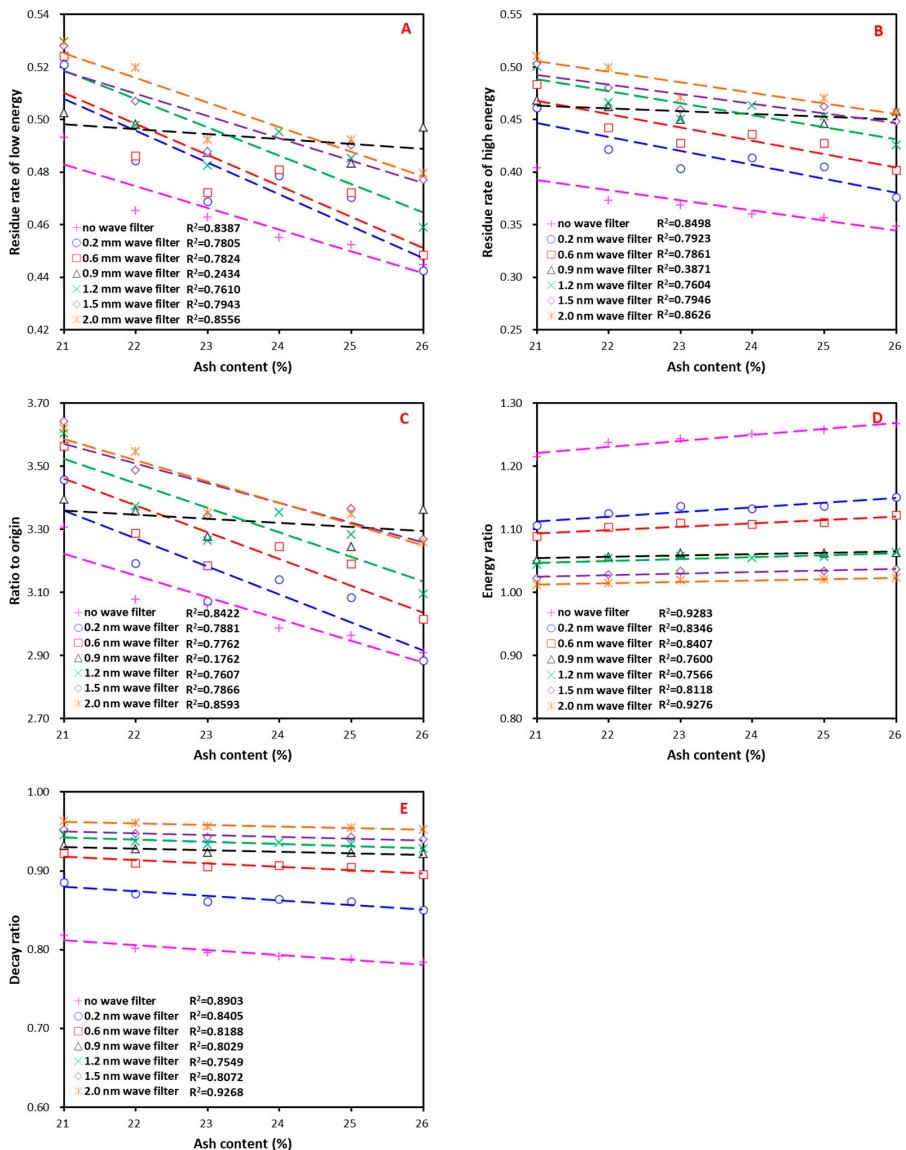

**Figure 4.** The relationship between ash content and characteristic parameters (**A**): residue rate of low energy; (**B**): residue rate of high energy; (**C**): ratio to origin; (**D**): energy ratio; (**E**): decay ratio with wave filter of different thickness.

### 3.2. Optimization of Tube Voltage and Tube Current

The adjustment of tube voltage and tube current would change the absolute intensity and energy distribution of the emitted X-ray, thus the tube voltage and tube current were also optimized.

The six samples in wave filter optimization experiments were still used in this section. The wave filter of 2 mm and single layer (50 mm) sample were used in the experiments. All experiments were repeated six times, and the average was reported. The results are shown in Figure 5. For energy ratio and decay ratio, the best correlation with ash content was found with tube voltage of 160 kV and tube current of 3125 vA, while for residue rate of low energy, residue rate of high energy, and ratio to origin, the best correlation with ash content was with tube voltage of 160 kV and tube current of 1250 vA. With comprehensive consideration of the five characteristic parameters, tube voltage of 120 kV and tube current of 1250 vA possessed the highest average linear coefficient factor, therefore this was the optimum working condition.

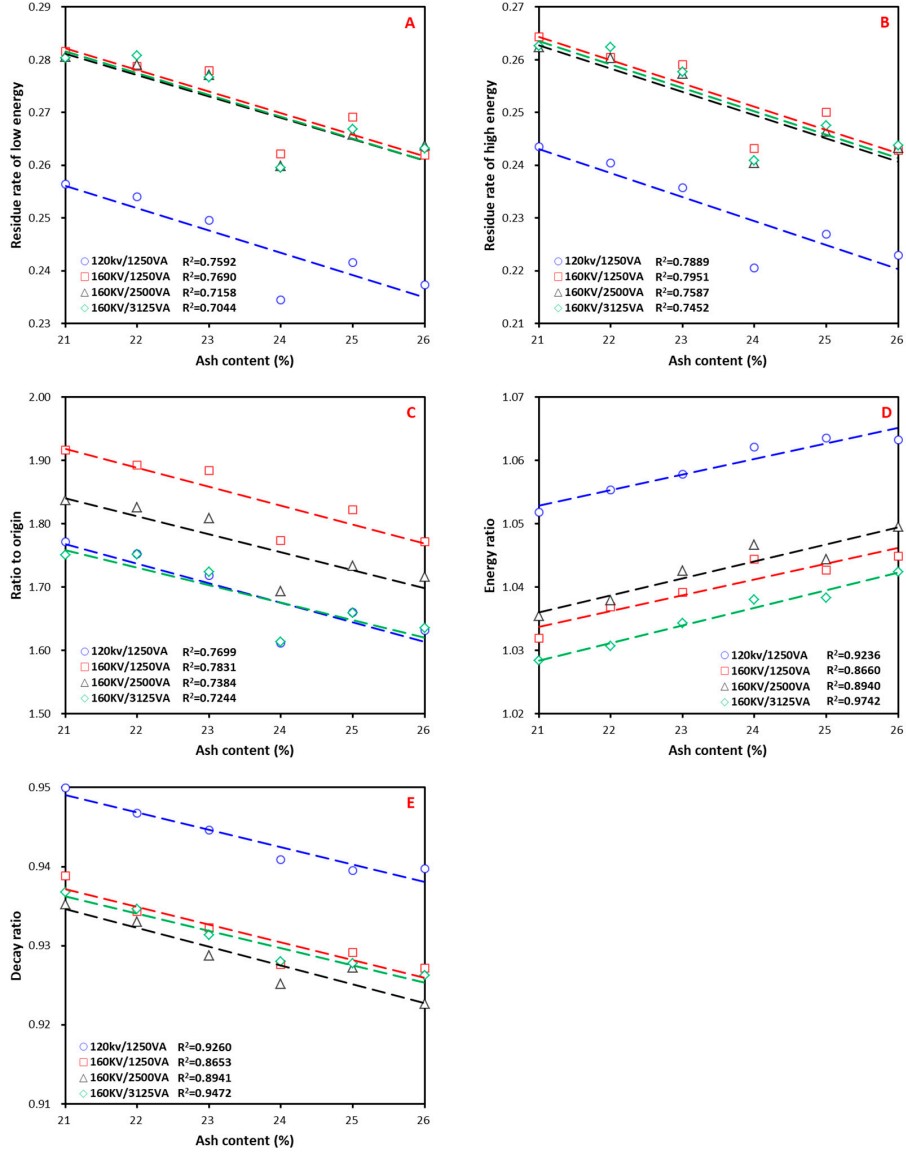

**Figure 5.** The relationship between ash content and characteristic parameters (**A**): residue rate of low energy; (**B**): residue rate of high energy; (**C**): ratio to origin; (**D**): energy ratio; (**E**): decay ratio at different tube voltages and tube currents.

### 3.3. Measurement of Wide Range Low-Moderate Ash Content

The coal samples of wide range low-moderate ash contents were obtained by mixing the coal (−6 mm) of different density fractions with predetermined ratios. The coal samples of 25 different ash contents ranged from 8.63% to 40% were prepared as single layer (50 mm), and then measured using the X-ray ash content analyzer. The tube voltage was 120 kV, the tube current was 1250 vA, and the pure nickel wave filter of 2 mm was used. During ash content calculation, permutations, and combinations of the five characteristic parameters were conducted. The five characteristic parameters were first normalized and then fitted linearly based on least square method. The ash content measured by the muffle furnace was correlated to the five parameters by 16 different algorithms. After comprehensive consideration of absolute error and coefficient factor, residue rate of high energy, energy ratio, and decay ratio were finally selected to calculate ash content.

The comparison of ash contents measured by the muffle furnace and the X-ray analyzer and the absolute error are shown in Figure 6. It was found that the maximum absolute error was 1.99%, while the minimum was −1.64%. The possibility of the absolute error smaller than 1.5% was 79.17%. The standard deviation was 0.98%. The possibility was respectively 66.67% and 95.83% for absolute error smaller than σ and smaller than 2σ.

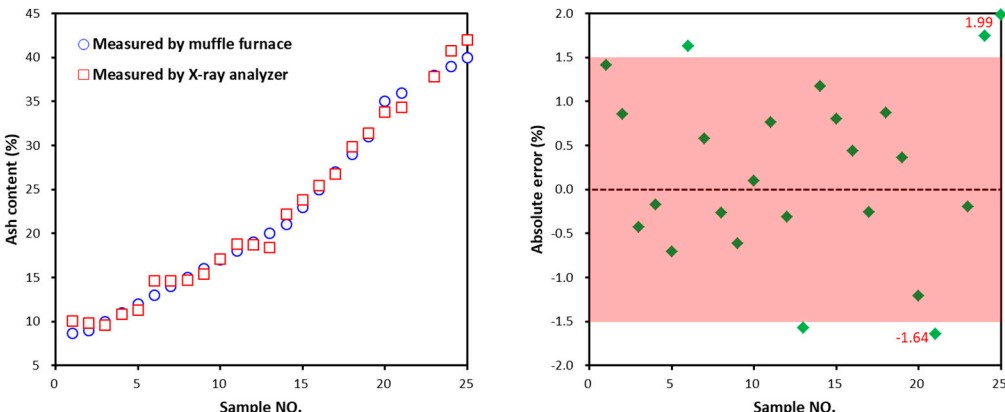

**Figure 6.** The difference (**left**) and absolute error (**right**) between the ash contents measured by muffle furnace and X-ray analyzer for slack coal with ash content between 8.63% and 40%.

### 3.4. Measurement of Narrow Range Low Ash Content

The coal samples of narrow range low ash contents were the clean coal (−6 mm) obtained from the float-and-sink experiments of the run-of-mine coal sample. The coal samples of 25 different ash contents ranged from 9.70% to 11.62% were prepared as single layer (50 mm), and then measured using the X-ray ash content analyzer. The tube voltage was 120 kV, tube current was 1250 vA, and the pure nickel wave filter of 2 mm was used. Residue rate of high energy, energy ratio, and decay ratio were still selected to calculate ash content.

The comparison of ash contents measured by the muffle furnace and the X-ray analyzer and the absolute error are shown in Figure 7. The maximum absolute error was 0.41%, while the minimum was −0.67%. The possibility of the absolute error smaller than 0.5% was 98.33%. The standard deviation was only 0.22%. The possibility was respectively 65% and 98.33% for absolute error smaller than σ and smaller than 2σ.

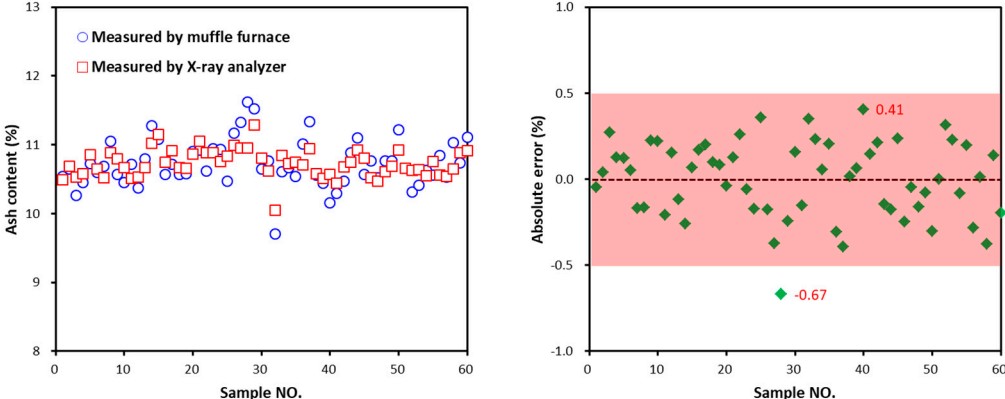

**Figure 7.** The difference (**left**) and absolute error (**right**) between the ash contents measured by muffle furnace and X-ray analyzer for industrial heavy medium clean coal with ash content between 9.20% and 11.15%.

Compared with other real-time ash content analyzers, the X-ray ash content analyzer based on pseudo-dual energy X-ray transmission has the following characteristics [1,6]:

1.  The detecting area of the X-ray ash content analyzer is a fan sector that can measure the whole fracture surface of the sample on belt, which is much wider than that of a γ-ray ash content analyzer that only measures a single point with diameter of 1–2 cm of the sample. This improves the sample representativeness and measurement accuracy, as well as simple control procedure;
2.  Compared with a passive ash content analyzer which measures ash content by the natural radioactive elements in coal, the X-ray ash content analyzer has higher ray intensity and signal to noise ratio;
3.  The measurement accuracy of the X-ray ash content analyzer is lower than that of a neutron activation ash content analyzer when there is large fluctuation in coal quality. The standard deviation of the X-ray ash content analyzer is around 0.4%, while it is around 0.3% for a neutron activation ash content analyzer. However, the X-ray ash content analyzer is easier to popularize in industry due to its lower price and simpler control procedure;
4.  Compared with an induced laser ash content analyzer, which may cause sparks on coal surface, the X-ray ash content analyzer is safer and more adaptable.

*3.5. Industrial Application of X-ray Ash Content Analyzer*

Based on the laboratory experiments, an industrial X-ray ash content analyzer was installed on a clean coal belt conveyor in a coal preparation in Shanxi province, China, as shown in Figure 8. After installation and shakedown test, the stable operation of ash content analyzer was achieved. The daily ash content measurement results from 21 July 2021 to 16 August 2021 with 191 points were plotted in Figure 9. The results show that the ash content measured by muffle furnace was from 9.20% to 11.15%. The absolute error between the ash contents measured by the muffle furnace and the X-ray ash content analyzer was within 1%, with a maximum of 0.68% and a minimum of −0.75%. The possibility of the absolute error smaller than 0.5% was 86.39%, and the standard deviation was 0.34%. The possibility was respectively 60.73% and 96.86% for absolute error smaller than σ and smaller than 2σ.

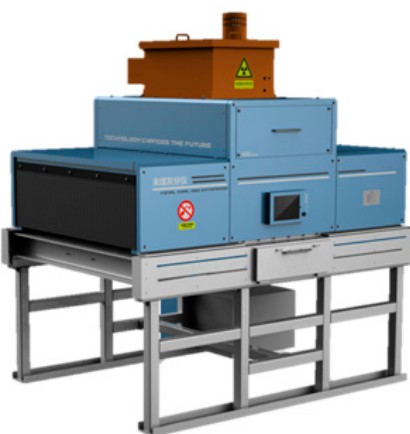

**Figure 8.** Industrial X-ray coal ash content analyzer.

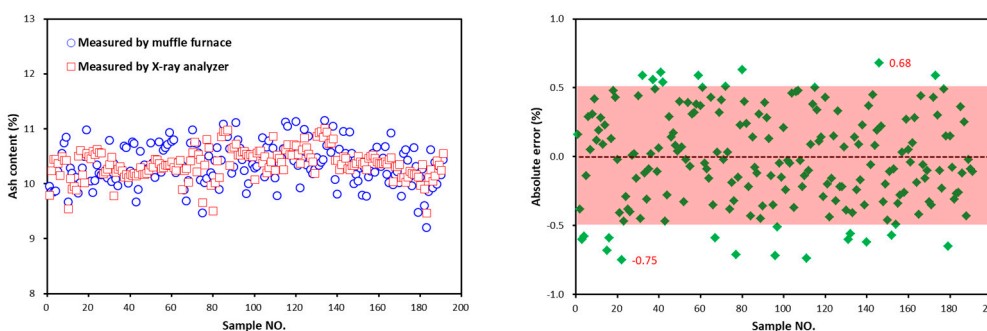

**Figure 9.** The difference (**left**) and absolute error (**right**) between the ash contents measured by the muffle furnace and the X-ray analyzer for industrial heavy medium clean coal with ash content between 9.20% and 11.15%.

The clean coal ash content requirement in this plant is between 10.50% and 11.00%. There would be a loss of clean coal recovery if the accrual ash content was lower than the requirement, and an impact on clean coal quality if the ash content was higher. The delay of the manually measured ash content by muffle furnace is difficult for the timely adjustment of operation parameters, such as the density and circulating pressure of heavy medium. There is a strong correlation between the measurement results of the X-ray ash content analyzer and the muffle furnace, and the variation trend is also consistent, therefore the X-ray ash content analyzer is capable for the real-time ash content measurement with acceptable error.

The comparison of Figures 7 and 9 indicates that the performance of the X-ray ash content analyzer was better in laboratory than in industry in respect of standard deviation and error distribution. This is mainly attributed to the larger fluctuation of coal quality, particle size, moisture, and belt material in industry. This fluctuation would influence the measurement accuracy [13]. Further investigation should proceed to improve the adaptability and precision of X-ray ash content analyzer.

## 4. Conclusions

The X-ray ash content analyzer based on pseudo-dual energy X-ray transmission is feasible for real-time ash content measurement. Five characteristic parameters, i.e., residue rate of low energy, residue rate of high energy, ratio to origin, energy ratio, and decay ratio, had good correlation with ash content, with a linear coefficient factor higher than 0.9.

The metal wave filter made of pure nickel could reduce the soft ray in X-ray beam and narrow the wavelength of X-ray. The optimum wave filter thickness was 2 mm. The optimum tube voltage and current was 120 kV and 1250 vA, respectively.

The laboratory and industrial ash content measurement results show that the pseudo-dual energy X-ray ash content analyzer is capable of real-time ash content measurement. For clean coal with ash content around 10%, the absolute error was within 1%, and the possibility of the absolute error smaller than 0.5% was higher than 85%.

**Author Contributions:** X.Z.; methodology, validation, investigation, formal analysis, writing—original draft preparation; L.L.; methodology, writing—review and editing, funding acquisition; T.L.; resources, writing—review and editing; J.T.; writing—review and editing, funding acquisition; X.L.; writing—review and editing; G.X.; writing—review and editing, supervision. All authors have read and agreed to the published version of the manuscript.

**Funding:** This research was funded by the Natural Science Foundation of Jiangsu Province (BK20200637).

**Acknowledgments:** Xiufeng Zhang would like to express thankfulness to her coworkers in Meiteng for their assistance in equipment installation and experiments.

**Conflicts of Interest:** The authors declare no conflict of interest.

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
