# Peer review of "Coal Ash Content Measurement Based on Pseudo-Dual Energy X-ray Transmission"

_minerals, doi:10.3390/min11121433_

Round 1

Reviewer 1 Report

This is a fine work that opens new ways for fast reliable ash content determination on-line directly in production facilities. It also appears to be safer that radiometric methods which usually include dangerous sources.

Some minor remarks I do have:

language:

line 53 embedded in coal

line 54 main drawback

line 88 which has to be safe

line 304 price

line 324 there is

content:

line 70 it needs to be 106

line 106 why did you use XPS? There seems to be no need for that and an explanation is not given. Rather, an ultimate analysis using elemental analyser would have been more useful.

line 122 where does the carbon content come from? XRF does not measure it. If it is from XPS then what is the meaning, since XPS is a strict surface method.

line 126 is it really possible to give results for optical measurements as accurate as X.XX %? This seems a really exaggerated precision.

line 151 which X-ray anode was used? Nowhere in the text any reference to that is given.

lines 268, 286, 317 this is standard mathematics and need not be cited here

Author Response

Reviewer #1:

Comment 1: Language.

Response: Thank you very much for your comments. We have corrected all the errors that you pointed out and checked through the language of the manuscript.

Comment 2: Line 70 it needs to be 106

Response: We have corrected this mistake.

Comment 3: Line 106 why did you use XPS? There seems to be no need for that and an explanation is not given. Rather, an ultimate analysis using elemental analyser would have been more useful.

Response: Thank you very much for your suggestion. We have deleted the XPS result and replaced it by ultimate analysis, as attached below, or you can see it in Table 2 in Page 3 of the revised manuscript.

Table 2. The ultimate analysis of the clean coal sample (air dried basis)

Element

C

H

O

N

S

Concentration (%)

81.32

5.22

10.59

2.06

0.81

Comment 4: Line 122 where does the carbon content come from? XRF does not measure it. If it is from XPS then what is the meaning, since XPS is a strict surface method.

Response: Thank you for your concern. We have interpreted the carbon content in Table 3 at the end of the first paragraph in Page 3.

The carbon element in Table 3 represents the light element such as H, C, N, etc., which is also called combustible matter, mainly composed of the organic matter and crystal water in the sample.

Comment 5: Line 126 is it really possible to give results for optical measurements as accurate as X.XX %? This seems a really exaggerated precision.

Response: We have changed the precision of the maceral analysis. You can see it in the last paragraph of Page 3 in the revised manuscript, or as attached below:

The proportion of vitrinite was between 34% and 56% and averaged at 41%, while it was between 24% and 42% and averaged at 36% for inertinite.

Comment 6: Line 151 which X-ray anode was used? Nowhere in the text any reference to that is given.

Response: We have added the details of the X-ray anode that we used, as well as reference [30] in the revised manuscript. You can see it in the last paragraph in Page 5, or as attached below.

A VJ X-ray tube was used in this work. The cathode filament and anode target were both made of tungsten. The aim to use tungsten is because it can release X-ray of high energy and strong penetrability after bombardment [30].

Comment 7: Lines 268, 286, 317 this is standard mathematics and need not be cited here.

Response: Thank you for your suggestion. We have deleted Eq. (2) and corresponding description.

Reviewer 2 Report

Manuscript ID: minerals-1506593

Title: Coal ash content measurement based on pseudo-dual energy X-ray transmission

Authors: Xiufeng Zhangуе et al.

Introduction. Line 68. What authors mean “expensive and complicated in operation”? Can the authors provide a comparison of cost and complexity?

Line 80. Change X-ray image to X-ray pattern. Use it in all article text.

Line 82-89. Authors repeat information from lines 29-81. This text can be removed.

Line 94-95. Add reference for this information.

Figure 1. The scale bar is very hard to see.

Section 2.2. What is the time of X-ray analysis for one detection?

Figure 3. Why the correlation on the Double layer is always lower compared to the Single layer. Authors must write about this fact.

Line 219-229. Why do authors use so similar samples (22-26% of ash content). Why authors didn’t use 20, 30, 40, 50, 60, 70, 80% of ash content? The same question is for Section 3.3.

Line 294. How much wider is the detecting area of X-ray compared to γ-ray?

Line 301. Write the comparison of “the measurement accuracy” in %.

Author Response

Comment 1: Introduction. Line 68. What authors mean “expensive and complicated in operation”? Can the authors provide a comparison of cost and complexity?

Response: We have added detailed information of the price and complexity of the neutron activation and laser-induced ash content analyzer in the end of the third paragraph in Page 2 in the revised manuscript, as attached below.

The price of neutron activation and laser-induced ash content analyzers are respectively higher than $ 780, 000 and $390, 000. It is only around $160, 000 for an X-ray ash content analyzer. The neutron activation ash content analyzer includes radioactive reactor that requires strict control procedures and special maintenance, while laser-induced ash content analyzer is usually coupled with sampling and preparation system and other auxiliary equipment.

Comment 2: Line 80. Change X-ray image to X-ray pattern. Use it in all article text.

Response: Thank you for your suggestion. We have revised this word in the manuscript.

Comment 3: Line 82-89. Authors repeat information from lines 29-81. This text can be removed.

Response: Thank you for your suggestion. We removed this paragraph.

Comment 4: Line 94-95. Add reference for this information.

Response: We have added references 28 and 29 for this information.

  1. Wu G. TDS Ray Identification and Protection Research of Ha’erwusu. Coal Technology, 2018, 37(12): 323-326. (In Chinese)
  2. Wang X, Li L, Jia Q. Application of CXR-1000 X-ray raw coal sorting system in coal mine. China Coal, 2019, 45(1): 89-93. (In Chinese)

Comment 5: Figure 1. The scale bar is very hard to see.

Response: We have enlarged the scale bar in Figure 1, as attached below.

Comment 6: Section 2.2. What is the time of X-ray analysis for one detection?

Response: We have added the detailed signal collecting time and ash content detection time of the X-ray ash content analyzer in the first paragraph in Page 5, as attached below.

The signal collecting interval of the linear array detector system was in 1~10 ms. For actual ash content measurement, the signal collected in every second was divided into 8 frames to calculate the data package of ash content. The data package in every minute was averaged and reported to the user.

Comment 7: Why the correlation on the Double layer is always lower compared to the Single layer. Authors must write about this fact.

Response: We have added the interpretation of the difference between the results of double layer and single layer. Please see it in the last paragraph in Page 5 and the first paragraph in Page 6 in the revised manuscript, or as attached below.

Meanwhile, the results of single layer (50 mm) were higher than that of double layer (100 mm). This is because the sample of double layer adsorbs more energy than that of single layer. For double layer, more than 90% of both the low energy and high energy would be adsorbed by the sample when the ash content is higher than 60%, so the energy that can be detected after penetration is very weak and may cause data distortion. For single layer, the residue rate of energy is much higher, so more accurate measurement could be achieved. With numerous exploiting experiments, it was found that the satisfactory accuracy can be obtained when the residue rate of high energy is higher than 30%.

Comment 8: Line 219-229. Why do authors use so similar samples (22-26% of ash content). Why authors didn’t use 20, 30, 40, 50, 60, 70, 80% of ash content? The same question is for Section 3.3.

Response: We have added the reason why we chose samples of similar ash content for the experiments. You can see it in the first paragraph in Page 7 in the revised manuscript, or as attached below.

The reason why the samples with similar ash content were used rather than that with sharp difference is that the correlation between characteristic parameters and ash content is usually worse when the ash content difference is smaller. If good correlation can be found under this condition, then better results can certainly be obtained when the ash content difference is large. On the other hand, the ash content analyzer is used to measure single sample in actual production process, such as run-of-mine coal, clean coal, or middlings, the ash content fluctuation of these single samples would not very large and is usually within 5%.

Comment 9: Line 294. How much wider is the detecting area of X-ray compared to γ-ray?

Response: We have added the detained information of the detecting area of X-ray and γ-ray in the third paragraph in Page 10 in the revised manuscript, as attached below.

1) The detecting area of X-ray ash content analyzer is a fan sector that can measure the whole fracture surface of the sample on belt, which is much wider than that of γ-ray ash content analyzer that only measures a single point with diameter of 1-2 cm of the sample. This improves the sample representativeness and measurement accuracy, as well as simple control procedure.

Comment 10: Write the comparison of “the measurement accuracy” in %

Response: Thank you for your suggestion. We have added the quantitative comparison of the measurement accuracy in fifth paragraph in Page 10 in the revised manuscript, as attached below.

3) The measurement accuracy of X-ray ash content analyzer is lower than that of neutron activation ash content analyzer when there is large fluctuation in coal quality. The standard deviation of X-ray ash content analyzer is around 0.4% while it is around 0.3% of neutron activation ash content analyzer. But the X-ray ash content analyzer is easier to popularize in industry due to its lower price and simpler control procedure.

Round 2

Reviewer 2 Report

The authors have made a major revision to the article.

It can be accepted in the present form.